# Implementation of Technical and Technological Progress in Dairy Production

**Marek Gaworski**

Department of Production Engineering, Institute of Mechanical Engineering,
Warsaw University of Life Sciences—SGGW, 02-787 Warsaw, Poland; marek_gaworski@sggw.edu.pl;
Tel.: +48-22-593-4583

**Abstract:** The involvement of people and technical devices is a characteristic feature of technological processes in agriculture. Human access to modernized and more efficient technical equipment determines the differentiation of the proportions of the contributions of human labor and technical equipment to the implementation of production technology on farms. Taking into account the data on manual and machine work inputs, the methodology of determining the technological index level ($T_L$) was presented. The aim of the present study was to present the scope of use of the technological index level to assess the effects of technological progress in the dairy production system, with particular emphasis on cow milking. For the value range of the technological index level (0–100%), changes in the milkman's work efficiency were presented based on research carried out on farms equipped with milking equipment at different levels of technical advancement. Moreover, the course of changes in electricity and water consumption per liter of milk was determined in association with the technological index level. The issue of simultaneous implementation of various forms of progress was developed based on the example of milking cows with a milking robot. Five categories (ranges) of cows' milk yield were distinguished and compared with the current yields of cows in the European Union. On this basis, a discussion was initiated on the factors that facilitate and limit the implementation of technical and technological progress in dairy production.

**Keywords:** dairy; engineering; farm; labor; process; progress

## 1. Introduction

Taking up and implementing production processes in agriculture is a key element in securing stable access to food products for the growing global population [1]. Hence, production processes in agriculture generate a wide spectrum of practical knowledge and scientific interests related to the improvement of food production. Improvement is a feature that distinguishes each area of agricultural activity, and dairy production in particular [2]. This is because milk plays a special role in feeding the population as one of the basic sources of protein, fat, and other nutrients [3–5]. Dairy production is an important area for the conversion of plant feed to human edible animal products in the food supply chain [6]. For many farms, milk production is an important source of income, which varies depending on biological, technical, and technological factors, as well as the market and the socioeconomic environment [7].

Providing consumers with access to high-quality milk and its products depends on the creation of an efficiently organized, sustainable dairy production system (DPS), reflecting the systemic approach identified in the fields of agriculture [8,9] and food economy. In a general sense, the dairy production system is a semantic description of an autonomous set of elements of a social, natural, and technical nature, synergistically interacting with one another [10] in order to transform various forms of energy and environmental components into products suitable for metabolic processing in living organisms [11].

The large number of objects and the relationships between them in the dairy production system (DPS) inspire the development of modeling research. The dairy production

system and its evaluation are sustainability issues encompassing three pillars, i.e., economic, environmental, and social [12]. The growing importance attached to the economic profitability, social welfare, and environmental impact of dairy production systems is an incentive to search for research tools for modeling considerations [13]. The link between dairy production and sustainability and modeling is demonstrated, for example, by the SIMSDAIRY (sustainable and integrated management systems for dairy production) model [14]. A feature of this model is the study of various interactions between internal (management, genetics, and others) and external factors, such as environmental conditions. Another WLGP (weighted linear goal programming model for dairy farms) model examines the interdependencies between biophysical, economic, and social processes in dairy farms [15], taking into account the maintenance of sustainable development. In turn, the GAMEDE (global activity model for evaluation of the sustainability of dairy enterprises) model presents an approach based on the analysis of the stock-flow, taking into account the operations and management of the dairy farm, which affect the sustainable income and profitability of production [16]. The approach involving intermediate products in the activity of dairy farms was the basis for the development of a model using a non-parametric technique to measure various forms of productivity [17]. One of the stages of modeling dairy production—especially in the earlier period—was the creation of optimization models with a specific range of analyzed data. The purpose of modeling was, for example, to assess the relationship between cows' feeding costs and their milk yield and income from milk production [18]. Another empirical economic model was developed to assess the effects of asset replacement in dairy production [19]. Economic indicators—including farm profitability in connection with biological, technical, and physical processes—were considered in the stochastic budget simulation model of a dairy farm [20]. An example of an approach to modeling an entire dairy farm is the DairyWise model; this model was based on the simulation of technical, environmental, and financial processes in a dairy farm, taking into account the supporting role of the FeedSupply and DairyHerd models [21]. In the nonlinear optimization model developed for the needs of the New Zealand dairy production system, mass flows in the area of dairy cattle feeding were assumed to be a key factor in making management decisions [22]. The approach that takes into account the balancing of the available feed on a dairy farm—related to the selection of the best feed production strategy for a given dairy herd, as in the DAFOSYM model [23]—confirms the special role of feed in modeling dairy farm production. Developing alternative solutions in the field of cow feeding, a whole-farm model (WFM) was proposed with the dominant share of feeding cows on pasture [24]. In the overall assessment of farms included in the dairy production system, significant emphasis is placed on the impact of dairy production on the environment; hence, the models of ammonia emissions [25] and nitrogen flow [26] in dairy farms make an important contribution to the assessment of dairy production management. Thus, dairy production and its evaluation are part of the circular economy [27] as a source of potential threats to the environment. The essence of the LCA (life cycle assessment) method is not only the assessment of the final result of the technological process, but also the estimation and assessment of the consequences of the process for the environment.

A characteristic feature of the formulated models of dairy farms and dairy production systems is the selection of a set of parameters (indicators); these are measurable, calculated, and descriptive indicators, classified as economic, social, and environmental categories that make up the three pillars of sustainable dairy production [13]. In practice, the concept of technical indicators is also used [28], which generally cover a broad set of factors, depending on the scope of the analysis. A question can be asked: what is the premise for creating such indicators? The main premises for creating new research indicators, and using previously known ones, can be the specificity of the research conducted, the goal(s) set for them, and the possibility of making comparisons with the results of other studies.

The aim of the present study was to present the scope of use of the technological index level to assess the effects of technological progress in the dairy production system,

with particular emphasis on cow milking. A characteristic feature of technological processes in dairy production—and in agriculture in general—is the inclusion of technical facilities and people in their implementation. It is these two pillars—i.e., the operation of technical equipment, and the work of people—that are the essence of the technological index level.

In general, the problem of assessing the use of machines and tractors, as well as the effects of their implementation in agricultural production, has been discussed in studies for many years [29,30], and has been formulated in terms of mechanization. The question is how to use the scientific aspect of research on the mechanization of technological processes, which is so closely related to agricultural practice [31]. The present study is an attempt to use the technological index level to present the problems of implementing progress or, rather, various forms of progress in dairy production. In addition to technical progress in agriculture [32], detailed studies also consider the issues of technological and constructional progress [33]. The authors of [34] linked technological progress with energy consumption in agricultural production. Taking up the issues of progress in the present study requires the definition of individual categories of progress. Technological progress consists of replacing manpower (manual labor) with technical means of work (technical devices). Technical/constructional progress includes the implementation of new devices, as well as the modernization of previously used ones.

Overall, progress is a concept that expresses the quantitative and qualitative changes in the state of an area over a period of time. In the discussion on progress, several characteristic trends have emerged, emphasizing the complexity of classifying basic forms of progress and the need to search for indicators for their assessment.

## 2. Materials and Methods

### 2.1. Identification of the Technological Process in Dairy Production Systems

A characteristic feature of the dairy production system is the ability to distinguish technological processes and groups of flowing main streams, which include mass, energy, and information. These complementary and simultaneously permeating streams are the source of the internal diversity of dairy production systems. The goal of providing consumers with a wide variety of dairy products implies a gradual transformation of the milk stream over time. This transformation takes place under the control of a specific information resource, and with the participation of energy streams contributed by technical factors, as well as the human factor involved in technological processes (Figure 1), varying in their degree of complexity.

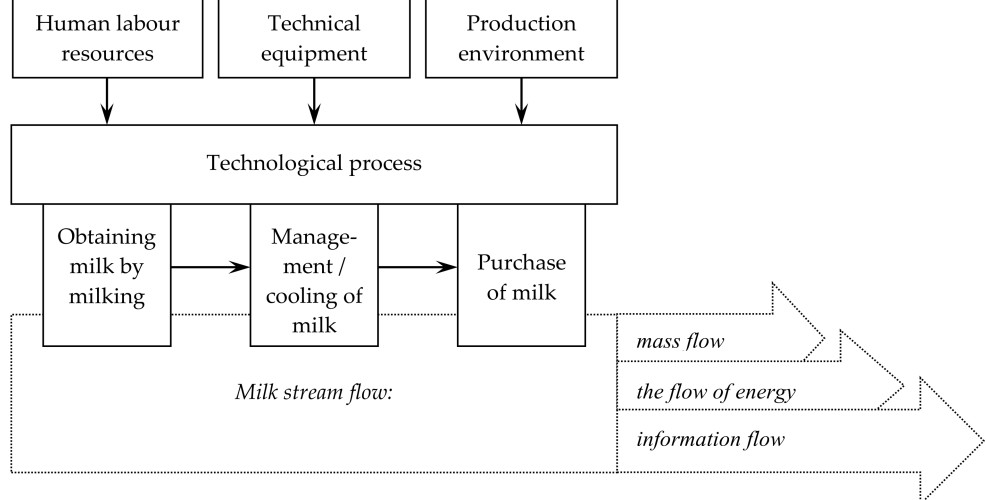

**Figure 1.** Technological process in dairy production systems, taking into account the group of flow streams and supporting human resources, as well as technical equipment and the production environment.

The diversity of the features characterizing individual objects and tasks in the processes of dairy production and management justifies the use of an appropriate research methodology [35]. Previous studies have distinguished many approaches to the analysis of dairy production and its diversification in terms of technical, technological, and other factors that make up the structure of the dairy production field. The technological approach, taking into account the possibility of using various organizational solutions in dairy production, creates conditions for making comparisons and selecting the most rational methods of operation using human and technical resources under the given economic and production conditions.

### 2.2. The Procedure for Calculating the Technological Index Level

The involvement of human resources and technical devices in the implementation of technological processes on farms has become a premise for the formulation of the term "technological level" [11]. The introduction of this concept to the analysis of a given research area implies the determination of the value of the technological level of a process or processes based on human work and technical devices. This value can be referred to as an index. Thus, the concept of the technological index level was used in our analysis.

The technological index level is calculated on the basis of the general procedure presented in Figure 2. In this procedure, the first step is to determine the recorded energy (Er) related to the performance of a given production process. The term "recorded energy" applies to human labor (E11r), work of tractors with machines or self-propelled machines (E12r), and machines powered by electric motors (E13r). Corresponding to the data collected, the recorded energy that reflects directly measurable inputs in the production process is expressed in conventional units—i.e., man-hours, kWh—which are used per unit (tons, liters) of harvested or processed plant or animal materials. The next step in the procedure (Figure 2) is the conversion of the recorded energy (Er) into usable energy (Eu). For this purpose, the energy utilization coefficients ($\beta 11$, $\beta 12$, and $\beta 13$) are used for human labor, machine/tractor work, and equipment powered by electric motors, respectively. Energy utilization coefficients ($\beta_{ij}$) are calculated on the basis of detailed methodological recommendations [36]. These coefficients take into account changes in the overall efficiency of working processes with the change in the degree of power utilization of energy sources, e.g., engines. In the case of manual work, the energy utilization coefficient ($\beta 11$), which is equivalent to the term "energy equivalent of working time" [37] used in the literature, is 1.96 MJ/man-h. As a result of the use of energy utilization coefficients in the calculation procedure, the usable energy (E11u, E12u, and E13u) is expressed in MJ units. Due to the expression in the same units, the usable energy components can be used to calculate the technological index level ($T_L$). The general formula for determining the $T_L$ index is the final step in the algorithm shown in Figure 2.

ENTRY TO THE COMPUTATIONAL ALGORITHM

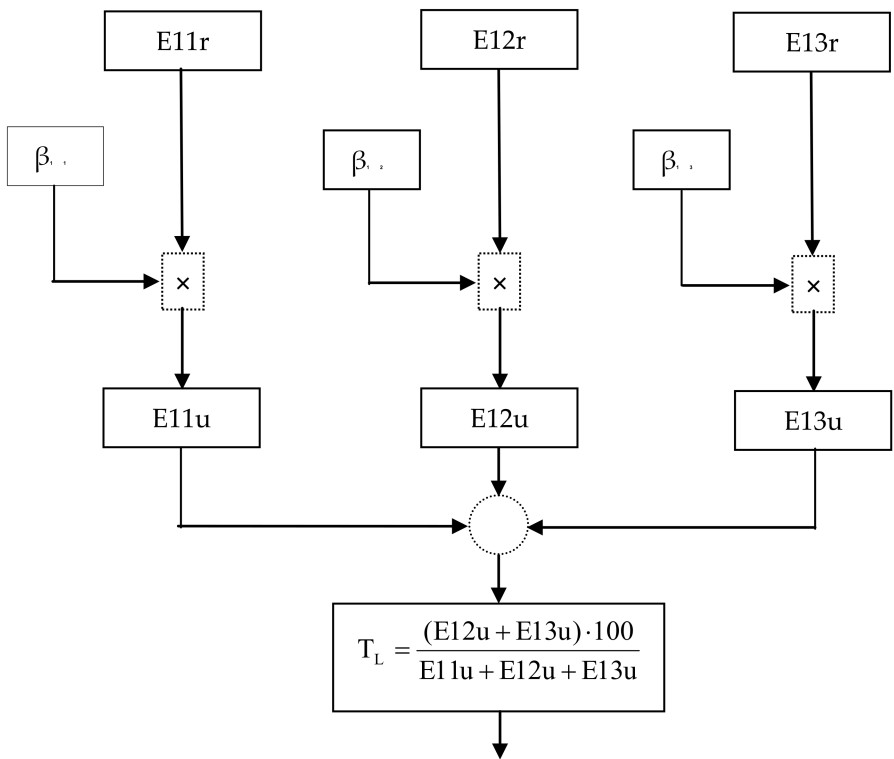

**Figure 2.** Algorithm to calculate the technological index level ($T_L$), where E11r (man-h/ref. unit): recorded human labor; E12r (kWh/ref. unit): recorded machine/tractor work; E13r (kWh/ref. unit): recorded energy of electrically powered devices; $\beta$11 (MJ/man-h): energy utilization coefficient for human labor; $\beta$12 (MJ/kWh): energy utilization coefficient for machine/tractor work; $\beta$13 (MJ/kWh): energy utilization coefficient for a device driven by an electric motor; E11u (MJ/ref. unit): usable human energy input; E12u (MJ/ref. unit): usable machine/tractor work energy input; E13u (MJ/ref. unit): usable energy input of electrical equipment; and ref. unit: reference unit.

The general formula for the calculation of the technological index level ($T_L$) presented in Figure 2 takes into account the case where, in addition to manual work energy, there is also the usable energy of work done by tractors/machines and electric-powered devices. In practice, the most common situations involve human labor and the work of machines/tractors. In the axiomatic evaluation system, the technological index level ($T_L$) is defined by the relationship between the share of machine/tractor work (E12u) in the total expenditure of human work (E11u), and machine/tractor work (E12u). This relationship is represented by the following formula:

$$T_L = \frac{E12u \cdot 100}{E11u + E12u}[\%] \tag{1}$$

where E11u is the usable human energy input, and E12u represents the machine/tractor work outlays in the form of usable energy.

In Formula (1), the usable energy expended by the machine or tractor takes into account the energy derived from the internal combustion engine. Energy can also come from animals, which in many farms around the world still provide tractive force for agricultural tools [38]. In practice, in farms—especially those involved in animal production—some

technical devices are powered by electric motors. Taking this into account, the formula for determining the technological index level may be given as follows:

$$T_L = \frac{E13u \cdot 100}{E11u + E13u} [\%] \tag{2}$$

where E11u—is the usable human energy input, and E13u is the usable energy from the electric motor.

The definition of the technological index level shows that its value covers the range $T_L \in\, <0, 100> (\%)$. A technological index level of $T_L = 0\%$ means that no machine works in the considered process. On the other hand, a $T_L$ index value of 100% indicates that the process does not involve personnel work, i.e., it is a fully automated process.

*2.3. Using the Technological Index Level to Analyze Technological Processes in the Dairy Production System*

The $T_L$ index can be widely used as a modeling variable describing the dynamics of changes in the considered parameters of the energy–technological state of objects or processes. This makes it possible to evaluate the obtained results of energy–technological analysis via national and international comparisons; it also allows for the assessment of structural changes in the studied area from a retrospective and prognostic perspective. The technological index level can be used to interpret many general phenomena in the field of food production, taking into account the use of human resources and technical potential under various production conditions [11].

The introduction to the analysis of the technological index level ($T_L$) is a premise for the unification and simultaneous systematization of detailed research studies of farms, diversified in terms of production potential. By systematizing the research area under consideration, Table 1 presents a set of key analysis concepts in connection with the values of the technological index level ($T_L$). The range of values of the technological index level (0–100%) can be divided into five equal parts, thus distinguishing five categories of a given concept. A farm is such a concept; hence, five categories of farms were distinguished, which differed in the value of the technological index level, resulting from the share of manual and machine work in technological processes in given farms.

**Table 1.** Key concepts of analysis in connection with the technological index level ($T_L$).

| Key Concepts in Analysis | | Technological Index Level—$T_L$ (%) | | | | |
|---|---|---|---|---|---|---|
| | Scope → | 0–20 | 21–40 | 41–60 | 61–80 | 81–100 |
| | Average → | 10 | 30 | 50 | 70 | 90 |
| Farm categories | | CI | CII | CIII | CIV | CV |
| Generations of agricultural machinery (technical infrastructure) | | $G_{am}I$ | $G_{am}II$ | $G_{am}III$ | $G_{am}IV$ | $G_{am}V$ |

In individual categories of farms, equipment representing a specific level of technical advancement is used. As a result of the use of this equipment in conjunction with human labor, the value of the technological index level is determined (based on labor and energy inputs). By replacing human work with machines, when the share of machines in the balance of inputs increases, the technological index level also grows. The increasing share of work carried out by machines is possible as a result of their technical advancement. Therefore, the aforementioned technical advancement may be the basis for distinguishing a generation of agricultural machines. As in the case of the five categories of farms, five generations of agricultural machinery/farm equipment were distinguished. Individual generations of devices differ in the technological index level values achieved as a result of their work.

The key concepts in the analysis—i.e., farm categories and generations of agricultural machinery—are summarized in Table 1. This summary includes the links between categories and generations with the technological index level ($T_L$), its five ranges, and average values for given ranges.

Each of the key elements of the analysis—i.e., farms, machinery/tractors, etc.—are characterized by a set of specific features. These are measurable, immeasurable, and in some cases also descriptive features. Examples of such features are listed in Table 2. In individual categories of farms characterized by the value of the technological index level, it is possible to identify measurable features related to plant production (arable land, grassland, yields, etc.) and animal production (herd size, annual milk yield per cow, etc.). This is similar in the case of the generation of agricultural equipment, which—apart from the demand for power—is also characterized by other parameters, both technical and functional (Table 2).

**Table 2.** Selected, measurable, and descriptive features of key concepts in the analysis of milk production on a farm, including elements of fodder and livestock production.

| Key Concepts in Analysis | Selected, Measurable (or Descriptive) Identification Features of Key Concepts |
|---|---|
| Farm categories | − Arable land area <br> − Grassland area <br> − Forage area <br> − Yield of forage plants <br> − Number of dairy cows in the herd <br> − Cows' milk yield <br> − Unit income from dairy production |
| Generations of agricultural machinery and equipment (technical infrastructure) | − Mass performance <br> − Operational efficiency <br> − Annual use <br> − Specific consumption of fuel, electricity <br> − Service qualification requirements <br> − The need for manual operation <br> − The degree of design modernity |

Selected, measurable features of the farm category (CI-CV) justify the selection and use of a specific generation of machinery and technical devices ($G_{am}I$-$G_{am}V$) for the implementation of specific tasks/technological processes. In the basic premise of the energy–technological analysis of agricultural production efficiency [11], the categories of farms are assigned equivalent generations of machines and technical devices, i.e., CI-$G_{am}I$, CII-$G_{am}II$, etc. (Table 3) Certain features of a given category of farms thus express the possible to apply the technical potential of agricultural machinery and equipment.

**Table 3.** Matrix of options for linking/adjusting the generation of agricultural technical equipment ($G_{am}$) with farm categories (C).

| | | | Farm Categories (C) | | | | |
|---|---|---|---|---|---|---|---|
| | | | CI | CII | CIII | CIV | CV |
| | | W [%] | 10 | 30 | 50 | 70 | 90 |
| Generation of technical equipment ($G_{am}$) | GV | 90 | $G_{am}V \leftrightarrow CI$ | $G_{am}V \leftrightarrow CII$ | $G_{am}V \leftrightarrow CIII$ | $G_{am}V \leftrightarrow CIV$ | $G_{am}V \leftrightarrow CV$ |
| | GIV | 70 | $G_{am}IV \leftrightarrow CI$ | $G_{am}IV \leftrightarrow CII$ | $G_{am}IV \leftrightarrow CIII$ | $G_{am}IV \leftrightarrow CIV$ | $G_{am}IV \leftrightarrow CV$ |
| | GIII | 50 | $G_{am}III \leftrightarrow CI$ | $G_{am}III \leftrightarrow CII$ | $G_{am}III \leftrightarrow CIII$ | $G_{am}III \leftrightarrow CIV$ | $G_{am}III \leftrightarrow CV$ |
| | GII | 30 | $G_{am}II \leftrightarrow CI$ | $G_{am}II \leftrightarrow CII$ | $G_{am}II \leftrightarrow CIII$ | $G_{am}II \leftrightarrow CIV$ | $G_{am}II \leftrightarrow CV$ |
| | GI | 10 | $G_{am}I \leftrightarrow CI$ | $G_{am}I \leftrightarrow CII$ | $G_{am}I \leftrightarrow CIII$ | $G_{am}I \leftrightarrow CIV$ | $G_{am}I \leftrightarrow CV$ |

The analysis of each of the linking options (Table 3) may be a premise for a detailed assessment of the effects of matching or mismatching a given generation of technical

devices to a given farm category. Individual categories (or generations) are identified by measurable features with a specific range of values. Hence, they can be confronted with one another, and conclusions can be drawn regarding the positive or negative effects of linking farm categories and their features with the generations of technical equipment.

### 2.4. Linking the Technological Level with the Assessment of the Implementation of Various Forms of Progress

The value of the technological level related to the implementation of tasks on the farm reflects the level of technical advancement of the agricultural equipment used. The transition from the lowest to the highest categories of farms is accompanied by the use of more and more modern technical equipment. The features of modern equipment are the main factor identifying technical progress. In practice, technical progress is implemented via increasingly higher generation equipment used on farms.

In addition to technical progress, other forms of progress can also be distinguished in the dairy farm. One such form is biological progress, as identified by the milk yield of cows showing a systematic upward trend over the past decades. Cows' milk yield is one of the measurable features characterizing the farm categories included in Table 2. Technical progress can be interrelated with biological progress in dairy farms, using milking robots. The following cases may accompany the simultaneous implementation of these two forms of progress:

- Biological progress is ahead of technical progress;
- Technical progress is ahead of biological progress;
- Biological progress and technological progress converge in terms of specific, measurable features.

Various forms of progress accompanying dairy production are an inspiration to evaluate the effects and consequences of their simultaneous implementation.

### 2.5. Research Materials in the Area of Dairy Production

The data needed to conduct an energy–technological analysis of milk production—with particular emphasis on milking—were collected during visits to 30 dairy farms located mainly in Poland. During visits to the farms, data on cow milking were collected. The power of the engines that powered the milking unit and vacuum pump, along with the compressor (if used) and the associated devices (washers), were identified. It was assumed that the engines were selected by the producers of individual milking installations, taking into account the same power reserves.

The data collected on farms included the number of cows in the herd, the milking time of the herd, and the amount of milk milked. The number of milkers was taken into account for the calculation of the milking efficiency indices. In each farm, observations were made for morning and afternoon milking (only farms with two milkings per day were visited). In farms using milking robots, the time of employee involvement, the amount of milk harvested during the day, and the number of cows in the herd were determined in order to calculate the performance evaluation indicators. The amount of water used in the milking and washing process was estimated for each of the farms.

In order to standardize the collection of data on farms, an appropriate questionnaire sheet was developed, taking into account the division of information into categories covering farms, barns, milking equipment, employees, and cows.

The visited farms used bucket and pipeline milking machines, tandem milking parlors, herringbone, side-by-side, and carousel milking parlors, as well as single- and double-stall milking robots; farms keeping a few hand-milked cows were also visited. The size of the dairy cow herds in the farms visited ranged from 2 to 285 cows. The age of the cows and their lactation were collected, but the data were not included in the calculations.

The data collected on the farms were used to calculate the technological index level ($T_L$), and in the next stage, in order to present the changes of the considered cow milking indicators along with the increase in $T_L$, the Statistica v.13 program [39] was used to plot

the course of changes in the analyzed parameters. The course of changes in variables was approximated by a linear fit, taking into account the regression interval with the forecast of 0.95.

## 3. Results and Discussion

### 3.1. Milkman's Work Productivity

The milkman's work productivity is one of the key indicators for assessing the efficiency of the milk production process in a dairy farm. The number of animals milked and the amount milked per hour play important roles in the evaluation of the milkman's work productivity.

Figure 3 shows the changes in the milkman's work productivity ($P_c$), expressed in terms of the number of cows handled per hour (c/m-h). In the analyzed range of variability of the technological index level ($T_L$), from 10% to 90%, an over 20-fold increase in the number of animals that can be handled during one man-hour was observed.

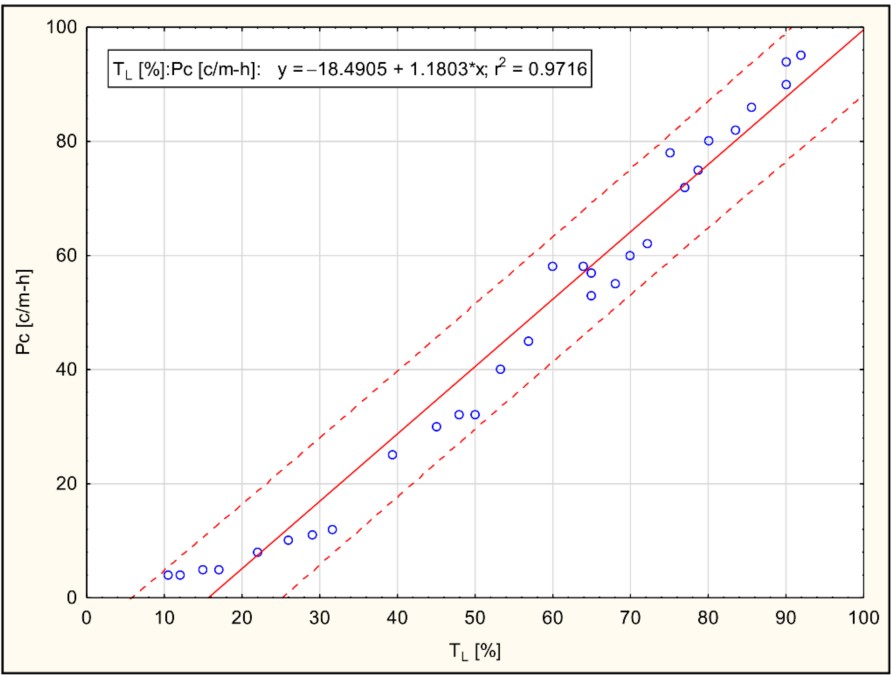

**Figure 3.** Changes in the milkman's work productivity ($P_c$) with an increase in the technological index level ($T_L$).

Reducing the manual labor input in the milking process is accompanied by a gradual re-evaluation of the role of employees, as well as growing requirements for their qualifications. These are requirements related to the ability to operate and supervise the increasingly complex and technically sophisticated devices that are standard equipment in modern milking parlors. The implementation of technical progress in the cow milking process implies the need for systematic improvement of the milkman's knowledge, which allows for full use of the functions of the milking equipment/technical systems on the farm.

Extending the scope of the analysis, Figure 4 shows the changes in the milkman's mass productivity, as expressed in the amount milked per hour by an employee. The achieved efficiency of the milking process (in kg/man-h), in addition to the type of technical equipment used, is also influenced by other factors, such as the milk yield of the cows and the intensity of milk flow from their udders.

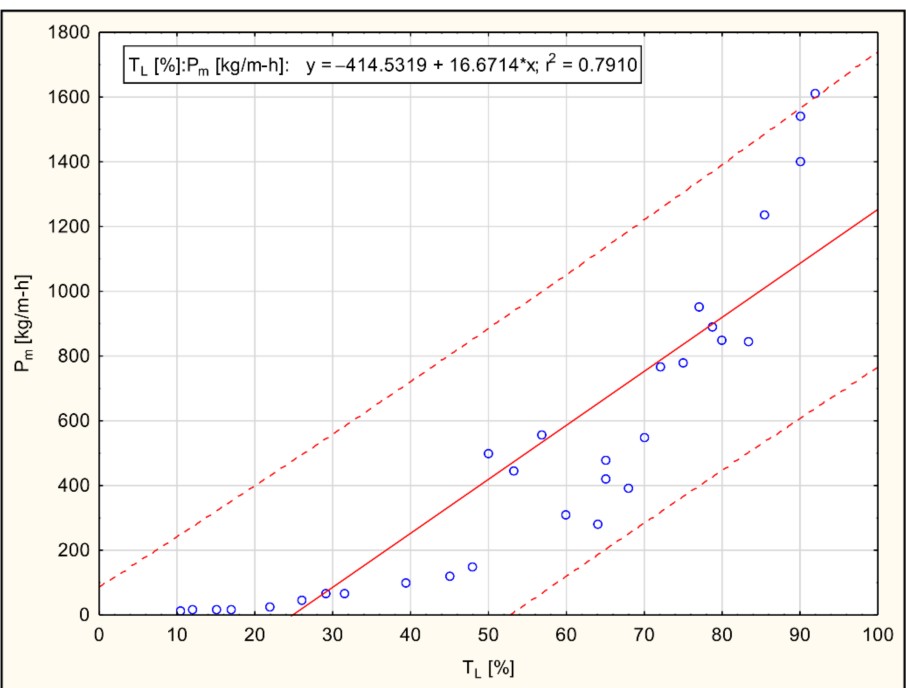

**Figure 4.** Changes in the milkman's mass productivity ($P_m$) with an increase in the technological index level ($T_L$).

The increase in the technological index level ($T_L$) in the range of 10–90% is accompanied by a more than 50-fold increase in the amount milked by the worker per hour, from ~20 to ~1100 L per man-hour. This highlights the benefits of using more and more modern technical equipment in the cow milking process. The increasing technical sophistication of the milking systems reflects technical progress. Examples of milking equipment representing the latest generation of equipment used on dairy farms include automatic milking systems (AMSs) and automatic milking rotary (AMR).

In addition to the number of cows and the amount of milk per milkman per hour, other indicators can also be used to analyze and compare various milking solutions. These include the milking efficiency per milking stall—including the number of cows per milking stall (heads/stall) and the amount of milk flowing through one milking stall (L/stall)—as well as the stall load index [40]. However, such indicators are mainly used in the case of milking parlors, representing a more advanced generation of technical equipment.

The use of more and more modern technical devices in the process of milking cows on farms translates into changes in the demand for power to drive vacuum pumps, milk pumps, and other working units. In milking equipment with increasing throughput of cows, engines with ever greater power are installed; this increases the potential power resources involved in the milking process. Electricity consumption is also changing. At the same time, increasing milking efficiency—expressed in the number of animals handled per hour—is accompanied by an increase in the amount of harvested milk. Figure 5 shows the changes in energy consumption in the cow milking process. An increase in the technological index level ($T_L$) in the range of 30–90% is accompanied by a decrease of ~30% in energy consumption per liter of milk.

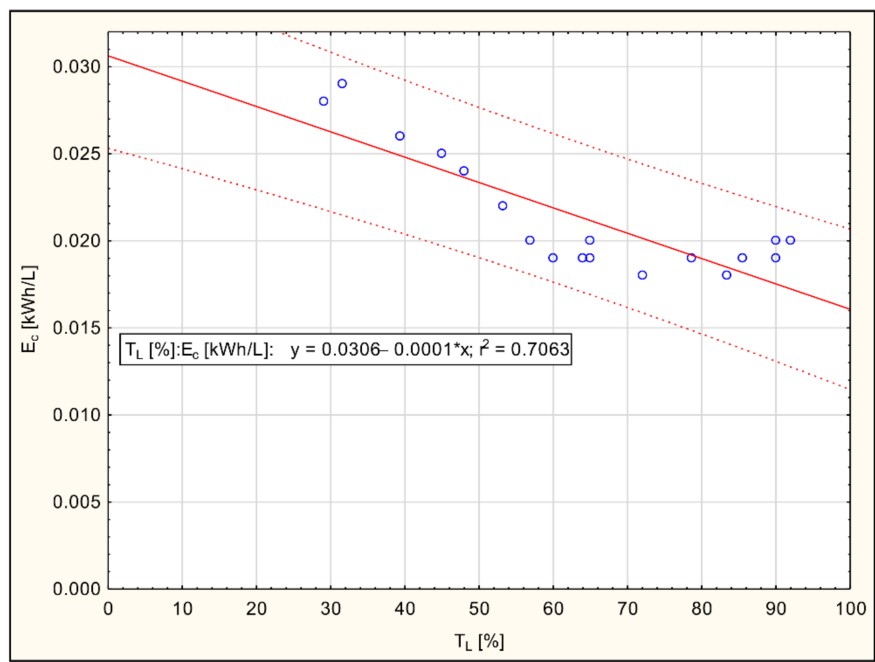

**Figure 5.** Changes in energy consumption ($E_c$) in the cow milking process with an increase in the technological index level ($T_L$).

The unit consumption of electricity (per liter of milk) in the range of higher values of the technological index level—represented by milking robots—is consistent with the results of [41]; in this study, carried out on farms with one and two milking robots, energy consumption ranged from 2.00 to 2.06 kWh per 100 L of milk. In [42], the total electricity consumption in farms with automatic milking systems was 62.6 Wh/L, of which 33% related to the operation of milking equipment. It can be calculated that such electricity consumption was at the level shown in Figure 5 for a $T_L$ index of ~90%, which corresponds to the use of AMSs.

Figure 6 shows changes in water consumption per unit of milk, in $L_w/L_m$ (liters of water per liter of milk). An increase in the technological index level ($T_L$) in the range of 10–90% corresponds to a reduction in water consumption, from ~0.5 $L_w/L_m$ (in the case of objects representing the lowest technological level) to ~0.22 $L_w/L_m$ (on farms of the highest category). It can be seen that for the $T_L$ range of 60–90%, the amount of water used in the milking process per one liter of milk is at a similar level. The differences in water consumption for milking between conventional milking parlors and milking robots in the test farm were confirmed by [43]. The assessment of water consumption in farms with milking robots was also undertaken by [42], who pointed to the differences in water consumption depending on the time of day, as well as in the following months of the year. The second indicator is related to the use of milking robots when cows have access to the pasture. Examples of these studies show that water consumption is determined by many factors, including those related to herd management.

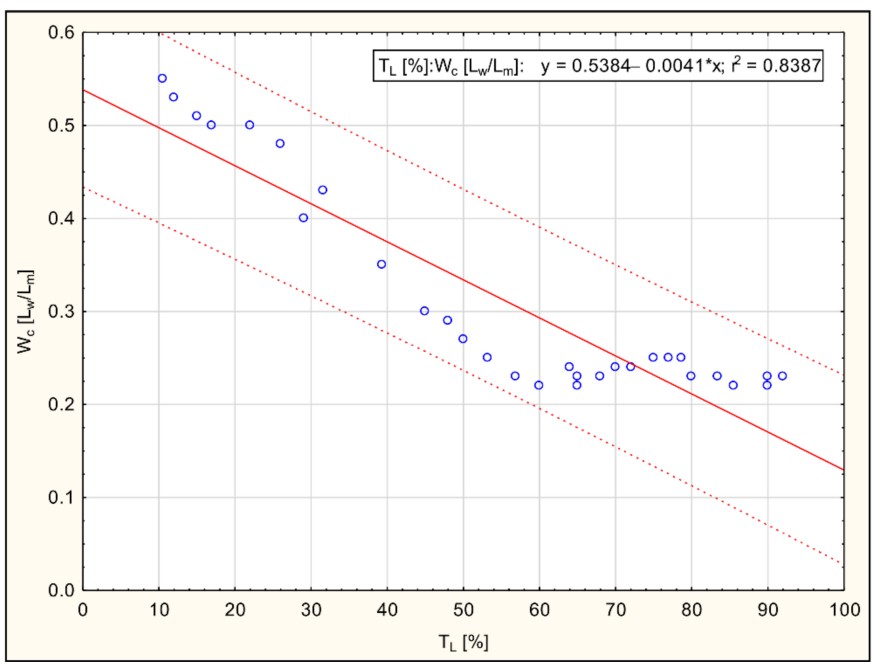

**Figure 6.** Changes in water consumption ($W_c$) in the cow milking process with an increase in the technological index level ($T_L$).

### 3.2. Simultaneous Implementation of Progress in the Field of Cow Milking

The issue of simultaneous implementation of various forms of progress was developed based on the example of the use of automatic milking systems (AMSs).

The automatic milking system represents the GV generation of technical equipment (Table 3). This next-generation equipment is associated with a herd of dairy cows, which can vary significantly in milk yield. The milk yield of cows represents biological progress, highlighting the changes in the animal's productivity over time. The milk yield of cows is one of the measurable features of the key concept in this analysis, i.e., the category of farm (Table 2). The five farm categories included in the study can be interpreted through the five categories of cows' milk yield. The establishment of the approximate range of variation in the milk yield of cows remains an open question. The milk yield of cows is one of the characteristic features identifying the production potential of dairy farms at the regional, national, and global levels. FAO statistical data [44] concerning regions of the world—including continents—were selected to determine the scope of milk yield differentiation. Table 4 summarizes the milk yields achieved in various geographic regions in 2019.

**Table 4.** Milk yields achieved in geographic regions in 2019 [44].

| Geographic Region | Milk Yield of Cows (kg cow$^{-1}$ year$^{-1}$) |
| --- | --- |
| Africa | 540 |
| Asia | 1956 |
| South America | 2953 |
| Oceania | 4571 |
| Australia and New Zealand | 4596 |
| Europe | 6498 |
| North America | 10,479 |
| World | 2699 |

Based on the data in Table 4, five categories of cow milk yield can be proposed, taking into account the range and the mean value for the range (Table 5).

**Table 5.** Categories ($C_{my}$) of milk yield of cows, including data expressed in kg cow$^{-1}$ year$^{-1}$.

| Category | $C_{my}I$ | $C_{my}II$ | $C_{my}III$ | $C_{my}IV$ | $C_{my}V$ |
|---|---|---|---|---|---|
| Range | 500–2500 | 2501–4500 | 4501–6500 | 6501–8500 | 8501–10,500 |
| Mean | 1500 | 3500 | 5500 | 7500 | 9500 |

Taking into account the criterion of the category of cows' milk yield, five adequate categories of dairy herds can be proposed.

Based on the proposed categories of cows' milk yield (Table 5), it is possible to assess the convergence with the generation of milking equipment represented by milking robots in selected countries. Based on the example of the European Union countries, the issue of simultaneous implementation of technical and biological progress can be raised. In other words, in such countries, the fifth generation of equipment ($G_{am}V$) corresponds to the fifth category ($C_{my}V$) of the milk yield of cows. The results, based on the 2019 FAO data [44], are presented in Table 6.

**Table 6.** Assessment of convergence of the implementation of the fifth generation ($G_{am}V$) of milking equipment with biological progress (milk yield of cows) in the European Union and United Kingdom [44].

| Link | $G_{am}V \leftrightarrow C_{my}I$ | $G_{am}V \leftrightarrow C_{my}II$ | $G_{am}V \leftrightarrow C_{my}III$ | $G_{am}V \leftrightarrow C_{my}IV$ | $G_{am}V \leftrightarrow C_{my}V$ |
|---|---|---|---|---|---|
| Country (milk yield of cows) | | Bulgaria (3627) Romania (3217) | Lithuania (6424) Slovenia (6178) Ireland (5783) Croatia (4608) | Portugal (8423) United Kingdom (8317) Germany (8246) Belgium (8088) Hungary (8077) Luxembourg (7780) Greece (7667) Austria (7215) Slovakia (7185) France (7153) Latvia (7073) Cyprus (6818) Malta (6744) Poland (6693) Italy (6661) | Denmark (9973) Estonia (9657) Spain (9178) Finland (9170) Netherlands (9154) Sweden (8973) Czech Republic (8731) |

The data presented in Table 6 show that seven countries are included in the highest category of cows' milk yield, four of which are Baltic states. Fifteen countries belong to the fourth largest category of cows' milk yield. There is no country in the first, lowest category of milk yield.

Regardless of the identification of countries in which the milk yield of cows is classified as convergent or non-convergent with the latest generation of milking equipment ($G_{am}V$), in practice it is also possible to indicate a different direction of the analysis. Detailed studies take into account the economic aspect of using milking robots, representing the latest generation of milking equipment.

A characteristic feature of the evaluation of milking robots in economic terms is the issue of not only the costs of equipping the farm with an AMS, but also the profitability of its use. One of the profitability indicators for the use of milking robots is the amount of milk per year. As this results from detailed analyses [45], the condition for the profitability of using a one-stall milking robot—as an example of implementing technical progress in a dairy farm—is the possibility of milking at least 500,000 L of milk per year. The same minimum economic profitability limit for using a milking robot is also indicated by other studies [46,47]. The authors of [48] give 600,000 L as the minimum annual amount of milk extracted by a one-stall milking robot.

The one-stall milking robot is adapted to handle a herd of 60–65 cows. Assuming 60 cows milked by the AMS, and 15% (in relation to the milked herd) dry cows, the total

number of cows on the farm can be 69 heads. Based on the amount of milk extracted per year that would guarantee the profitability of using the milking robot (600,000 L, i.e., 618,000 kg), along with the number of cows in the herd (69 heads), it can be calculated that the milk yield of one cow in a barn with an automatic system should be at least 8957 kg per year. It follows that the calculated milk yield of cows in a herd operated by a milking robot is in the range of the fifth milk yield category ($C_{my}V$). Thus, it can be seen that in any given case there is a convergence of the latest generation of milking equipment (AMSs) with the highest category of cows' milk yield. The economic analysis confirmed the need to select a herd of cows with the highest milk yield on a farm with an automatic milking system. The authors of [49], based on studies using the farm simulation model, found that the greatest benefits (farm net return) were obtained when AMSs were used at maximum milking efficiency; the greatest economic benefits come from milking robots that can milk a herd of 60 cows with a capacity of 8600 kg/cow.

The problem of simultaneous implementation of technical and biological progress in the field of cow milking is even more critically emphasized by other comparative data. When comparing the unit costs of milk production with conventional and automatic milking systems [50], it was found that these costs in the case of AMSs were higher as long as the amount of milk produced was less than ~800,000 kg per year. The number of cows in the herds operated by AMSs was on average 62 heads which, after taking into account a 15% rate of dry cows, gives just over 71 heads. It follows that in order to meet the requirements related to the profitability of using a one-stall milking robot, the milk yield of cows should be at least 11,268 kg per year.

The key issue in the present analysis of the minimum milk yield of cows milked by a robot is taking into account the number of animals in the herd. Based on research carried out on 197 farms using AMSs in Canada, the authors of [51] reported that the mean number of cows per AMS was $47.5 \pm 14.9$ (mean $\pm$ SD). Meanwhile, a study conducted on 41 AMS-equipped farms in Ontario and Alberta (Canada) [52] found that the average number of cows in a herd was 49. In other North American studies, the number of cows per AMS ranged from 51 [53] to 56 [54]. Based on modeling data from Galicia (Spain), the authors of [55] suggested the possibility of maximizing milk yield per AMS as a result of maintaining a large number of cows in the herd (e.g., 69 cows), with a reduced frequency of milking of individual animals. In [56], considering farms with 57 milking robots in Central Europe, the number of cows in the herd per AMS ranged from 47 to 69. In practice, it is recommended that the herd of dairy cows milked by AMSs should consist of approximately 60 cows [57]. This was confirmed by [58], who indicated that the widely accepted optimal herd stocking density is 60 cows/AMS. The discussions emphasize the complexity of the factors that determine the number of cows in the herd milked by an AMS, taking into account the effects of cow milk production, DIM (days in milk), and parity.

## 4. General Discussion

Increasingly modern technical equipment—i.e., tractors, machines, and other devices—constitutes the basis for the mechanization of farms and the technological processes carried out on these farms. Human labor resources and various available sources of power on farms are an inspiration to raise the issue of the status, challenges, and strategies of farm mechanization [59,60]. Modern and innovative concepts regarding dairy cattle production technology are closely related to the level of mechanization [61]. The concept of the level of agricultural mechanization appears in many studies—including those related, for example, to promoting the development of sustainable agriculture [62]. Another example of scientific considerations is enhancement of the level of agricultural mechanization in connection with the use of information technologies [63]. The extension of the approach to the agricultural mechanization level is its linkage with the level of agricultural equipment [64]. However, the word "level" in relation to agricultural mechanization is often used very generally, without giving a specific value. Summarizing the research on dairy farms, the authors of [65] emphasized the relationship between the suitable mechanization levels and the

reduction in the required amount of labor. The proposal to use the technological index level ($T_L$) in research is an attempt to quantify—i.e., quantitatively (in the form of a number or percentage value)—the phenomena presented descriptively, and related to the determination of the level of mechanization—especially when the description of the research uses the terms low, medium, or high level of mechanization. For example, the statement that a high level of mechanization in dairy farms determines both the possibility of achieving comfortable working conditions and the production of high-quality milk [61] requires clarification as to what is meant by the term "high level", and what is the line between high and medium levels of mechanization.

The essence of the technological index level is the comparison of mechanical work inputs with manual work inputs incurred in a given technological process. Of course, other factors, including descriptive ones—relating to, for example, various aspects of modernity—can also be selected for the assessment of production technology. Precision dairy technologies constitute one such example [66]. The purpose of implementing these technologies is to reduce the labor demand [67]. The demand for manual work is included in the formula for determining the technological index level, which thus confirms the usefulness of this index—for example to assess the modernity of technology, including dairy production technology.

In the discussions and research studies undertaken, attention should be drawn to the fact that one of the conditions for increasing the level of agricultural mechanization is the inclusion of IT elements in the construction, along with use of agricultural machinery [63]. The question remains as to how to accurately assess the impact of computerization on changes in the level of agricultural mechanization. The same question applies to the issue of innovative processes and their management [68], considered as an element of the assessment of technological progress.

The implementation of production technology in agriculture involves the consumption of various forms of energy. The present study of energy–technological analysis is an example of the use of labor and energy inputs to develop a technological index level. The essence of energy use in the assessment of production processes was confirmed by [69], who developed the "flow energy value map". This approach made it possible to plan production on a dairy farm, taking into account the criteria of the best economic, environmental, and social efficiency. The relationship between efficiency and energy consumption in dairy production systems is the subject of many studies, including those involving small dairy production systems [70]. These are studies showing the consumption of various forms of energy not only in dairy farms, but also in milk processing. The result of these and other studies [71] is to determine energy consumption (in MJ), as well as carbon dioxide emissions per kilogram of milk at the farm or processing stage. The energy consumption found in this research is the basis for looking for savings in dairy production and processing technologies. Addressing the issue of the energy intensity of milk production, the authors of [72] emphasized the need to formulate recommendations for the management of livestock production operations that will reduce energy consumption in dairy farms. The presented results of this research are based on a different approach. Information on the energy used was taken to calculate the technological index level ($T_L$) as an independent variable to evaluate the course of various parameters characterizing dairy production. The effect of using the technological index level was to indicate the possibility of introducing categories of farms differing in terms of production potential and other parameters. The proposed classification created in the agricultural production area on the basis of the $T_L$ index is consistent with the tendency to develop typological classifications in the dairy sector, which can be used, for example, to assess the sustainability of farms [73] and identify the differences in dairy management priorities [74].

The proposed classification of cows' milk yield based on five categories can be considered as an example of an approach to assess the differentiation of production potential on dairy farms. Every day this production potential is confronted on farms, in terms of the technical potential of the milking equipment. The result of this confrontation is

the achievement of specific milking efficiency indicators, including the milkman's work productivity, as well as energy and water consumption.

The present study indicates the essence of cows' milk yield in a herd operated by an automatic milking system. In farms using milking robots, it can be expected that cows with high (the highest) milk yield will be selected for the herd [75]. In practice, as shown by previous studies [76], farms with milking robots achieved higher milk yields compared to cows in farms using conventional (pipeline) milking systems. Similar trends were observed in other groups of researched farms; where milking parlors were used, the milk yield of cows was higher compared to herds on farms with bucket milking systems [77]. In other words, more sophisticated on-farm milking equipment can lead to higher milk yields [78]. Such a statement, however, requires confirmation in additional studies based on a larger number of dairy farms.

Another observation also requires confirmation with a larger number of dairy herds: Examples of analyses [79] indicate a tendency towards an increase in the milk yield of cows along with an increase in the number of cows in the herd. An increase in size of the herd justifies the selection of milking equipment with ever-greater milking capacity [80]. This is due to the transition to more and more advanced milking equipment. Thus, one of the observations presented in this study can be confirmed—that the latest generation of milking equipment ($G_{am}V$) may correspond to the highest category of milk yield ($C_{my}V$).

In previous studies [81], the production characteristics of dairy cows—including milk yield—were used for segmentation (grouping) of a herd operated by an automatic milking system. The evaluation of milk yield can therefore be considered not only in the context of different milking systems, but also within a single milking system (AMS).

The range of the considered milk yield of cows proposed in this study was established on the basis of data from 2019. The year of generation of the data for the analysis is important in this case, taking into account the tendency for the increase in the milk yield of cows over the years. For example, the research conducted in [82] compared the dynamics of the increase in cows' milk yield in three countries over two decades. In one country (Estonia), attention was paid to increasing the growth rate, while in the other two countries (Latvia and Poland), the focus was on reducing the growth rate of cows' milk yield.

The range of cows' milk yield proposed in this study was developed on the basis of data from various geographical regions of the world. The scope of the data can of course be limited to a selected region and, as a result, create a basis for comparing the milk yield of cows in that region.

The problem of differences in the milk yield of cows in comparison with the sophistication of milking equipment, as considered in this analysis, is part of the problem presented in the literature—referred to as the "yield gap". The concept of the yield gap—i.e., the difference between the theoretical maximum production and the current production of dairy cows—was the premise for the development of the dairy production systems analysis model [83].

The milk yield of cows is one of the key elements in the evaluation of dairy production on farms, and is the result of the breed and genetic potential of the animals, the specificity of the feeding system, including the type of feed, and the age, health status, and lactation period of the cow, among other factors [84]. A large number of factors relating to cows and the dairy production environment provide inspiration for forecasting the milk yield of cows, taking into account the herds milked via robotic systems. As a result of using appropriate statistical tools, including the decision tree technique [85], the most important factors responsible for the milk yield of cows can be identified. In the case of cows milked by AMSs, these factors were milking frequency, lactation number, and DIM (days in milk).

The problem of the diversification of agricultural production space has become a premise for attempts to classify agricultural facilities, including farms, their equipment, and agricultural production technologies. The creation of this type of classification allows the organization of the research area in order to, for example, adapt a development strategy for dairy production systems [86]. The question remains the choice of the farm classifica-

tion criteria and the number of distinguished groups of farms; in the mentioned studies, the latter consisted of six homogeneous groups of dairy farms. The research conducted by [87] defined three groups of farms in the typology of dairy production systems (DPSs), based on the assessment of economic, social, and environmental factors. As a result of the statistical analysis, the authors of [88] identified four groups of dairy farms differentiated by different levels of technology. The present study covers five categories of farms; size is the key criterion for distinguishing those five categories. Size refers to the characteristics under consideration, including area and number of animals (herd size). Moving to higher and higher categories of farms corresponds to an increase in the number of animals in the herd. The tendency towards an increasing share of dairy cows kept in larger herds was noted in [89]; however, the dynamics of this phenomenon vary between countries.

The effects of dairy farm enlargement were noted in [90], linking farm size with professionalization; the authors indicated that on larger farms, workers may be better paid, better trained, and may have more specialized skills; they can also achieve higher satisfaction [91] and work efficiency [92]. According to [93], with increasing farm size, the importance of manual (family) labor relatively decreases. The authors of [94] expressed the opinion that increasing the size of dairy farms is accompanied by a shift from family labor to external labor.

Human work in the dairy production system is subject to quantitative and qualitative assessment. As a result of the implementation of technical and technological progress, the quantitative human contribution to tasks in dairy farms decreases. Technical support for people translates to an increase in the quality of work performed, as well as its safety. Changing the proportion between quantitative and qualitative values in job evaluation should contribute to increasing the satisfaction of dairy farm owners in the process of transforming the food chain [95].

## 5. Conclusions

The technological index level can be considered as a tool for assessing production processes in agriculture, along with the effects of implementing technological progress. The effect of technological progress, taking into account the growing share of work inputs from technical devices in the total input of manual and machine work, is an increase in human work efficiency. In the case of cow milking, an increase in the technological index level in the range of 10–90% was accompanied by an approximately 20-fold increase in the number of cows milked. There was also an over 50-fold increase in the amount of milk per milkman. Technological progress, identified by an increase in the technological index level, is also expressed in a reduction in electricity and water consumption per liter of milk.

The essence of implementing technical progress in dairy production is to fully use the potential of modernity in the designed and modernized technical equipment. Rational use of the potential of milking robots, representing technical progress, is conditioned by the milking of a herd of dairy cows with an appropriate milk yield, representing biological progress. In this context, it is justified to develop further research showing the effects of the simultaneous implementation of various related forms of progress—not only in dairy production, but also in other areas of agricultural activity.

**Funding:** APC was partially funded by the Institute of Mechanical Engineering, Warsaw University of Life Sciences.

**Institutional Review Board Statement:** Not applicable.

**Informed Consent Statement:** Informed consent was obtained from all subjects involved in the study.

**Data Availability Statement:** Not applicable.

**Acknowledgments:** The author would like to thank all dairy farm owners for the opportunity to carry out the observations and measurements used to prepare the research study, as well as for all of the discussions that inspired him to consider the problem of evaluating dairy production systems.

**Conflicts of Interest:** The author declares no conflict of interest.

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
