# Peer review of "Implementation of Technical and Technological Progress in Dairy Production"

_processes, doi:10.3390/pr9122103_

Round 1
Reviewer 1 Report
The sentence in rows 497-499 "Introducing more and more modern technical equipment into agricultural practice, i.e. tractors, machines and other equipment, is identified in terms of mechanization of farms and the technological processes implemented in them" should be redefined.
English language and style are fine/minor spell check required.
Overall, future research should include more farms from other areas (not only Poland) and divide them by the dairy herds separated by variety so the numbers should be more precise.
Congrats, great work.
Author Response
Dear reviewer,
Thank you for preparing the article review and detailed comments.
The sentence in rows 497-499 "Introducing more and more modern technical equipment into agricultural practice, i.e. tractors, machines and other equipment, is identified in terms of mechanization of farms and the technological processes implemented in them" should be redefined.
I rewrote the indicated sentence to make it more understandable and meet the requirements of English language correctness.
English language and style are fine/minor spell check required.
I reviewed the article again for the correct use of the English language and made the appropriate corrections.
Overall, future research should include more farms from other areas (not only Poland) and divide them by the dairy herds separated by variety so the numbers should be more precise.
Thank you for your suggestions for further research. Future research will consider more foreign dairy farms and additional factors, including information on the quality of milk produced on farms.
All corrections have been made in red in the attached file with the article taking into account the changes and additions.

Reviewer 2 Report
An interesting scientific question
Chapter 2.2 should be rewritten, because it is currently difficult to understand. The important explanations are at the end and the formulas for which these explanations are needed are at the beginning. Furthermore, the source 11 is referred to, in which the system is explained. But this is not to be found for me (no DOI, no journal). A little more explanation would be good.
Line 151 ff: It is also not to be recognized which basis the system E11, E12 and E13 has. I can see which energies are called that, but I don't understand why.
Line 202 ff: The technology level refers to energy alone. The other two points "mass" and especially "information" are no longer considered. Why, since especially information is crucial in today's technology?
The two chapters 3.1 and 3.2 have little to do with each other except that they deal with milk and evaluate the technology. They use different material and different methods. 3.1 looks at Poland with the interaction of energy and labor in milk production. 3.2 analyzes AMS and herd performance in Europe, with an outlook on adjusted animal numbers per AMS. They are both interesting in their own right, but why do they need to be in one article?
The publication looks to me like it consists of several independent parts. The introduction and discussion are a good analysis of the literature on the subject. Material and method and chapter 3.1 are an interesting study on Polish dairy farms on energy and working time. Chapter 3.2 is a good analysis on milk yield and AMS and optimal herd size. However, I have not found the link that ties it all together into one publication.
Line 771: There is also a DOI here.
Line 773: Source misspelled. Correct: "Roboter: Zu viel Leerlauf."
The sources can be revised again altogether.
Author Response
An interesting scientific question
Dear reviewer,
I am very grateful for the preparation of the article review. The comments in the review were an incentive for me to think and to revise the way of presenting some of the issues raised in the article.
Chapter 2.2 should be rewritten, because it is currently difficult to understand. The important explanations are at the end and the formulas for which these explanations are needed are at the beginning. Furthermore, the source 11 is referred to, in which the system is explained. But this is not to be found for me (no DOI, no journal). A little more explanation would be good.
As suggested, I changed the structure of Chapter 2.2 to make the content presented in a more logical, clear and readable manner. I put the main emphasis on explaining the next steps in the calculation procedure leading to the determination of the TL index. The literature source marked with the number [11] is a book in Polish, published by the Polish Academy of Sciences. The author is my professor Tadeusz Nowacki (who died in 2005), whom I helped in the preparation of this publication. This study does not have a DOI number.
Line 151 ff: It is also not to be recognized which basis the system E11, E12 and E13 has. I can see which energies are called that, but I don't understand why.
In the general model of energy flow through the food economy system (based on the aforementioned publication [11]), five types of energy have been distinguished, i.e. environmental energy, direct energy, indirect energy, energy of produced products and energy losses. The energy forms E11, E12 and E13 are classified as direct energy. The full name of the form of energy (based on publication [11]) is: energy equivalent of work of personnel (E11), energy equivalent of work of machines (E12) and energy equivalent of work of electrical devices (E13). The presented considerations are complemented by the classification taking into account the levels of recorded energy (Er), usable energy (Eu) and cumulated energy (Ec). In the article I took into account the resulting concepts, i.e. E11r (recorded energy of human labour) etc. In the analysis I did not develop the assessment at the level of cumulated energy, because it is a more complex procedure.
Line 202 ff: The technology level refers to energy alone. The other two points "mass" and especially "information" are no longer considered. Why, since especially information is crucial in today's technology?
Thank you for your comment. I fully agree that information plays a key role in modern agricultural production (and not only). In dairy production, this would be information about the chemical properties of milk (milkfat and protein content), microbiological properties of milk (total bacteria counts - TBC, somatic cell counts - SCC), or the temperature of milk at various stages of its production, transport, processing and distribution, taking into account the fulfilment of the requirements for the cold chain. Thank you for highlighting this valuable aspect of the analysis that inspired me to develop further research. In this research, I will try to link the technological level (TL) of dairy production with the microbiological quality of milk and the maintenance of the cold chain.
The two chapters 3.1 and 3.2 have little to do with each other except that they deal with milk and evaluate the technology. They use different material and different methods. 3.1 looks at Poland with the interaction of energy and labor in milk production. 3.2 analyzes AMS and herd performance in Europe, with an outlook on adjusted animal numbers per AMS. They are both interesting in their own right, but why do they need to be in one article?
Thank you for paying attention to the content of sub-chapters 3.1 and 3.2. My intention was to present in the article aspects related to the implementation of technical and technological progress on the example of cow milking. In general, both subchapters present the results of research analyzes related to the milking of cows. The milking of cows and its evaluation is thus the central thread of the research considerations undertaken in both subchapters. In subsection 3.1, the evaluation of cow milking was developed, taking into account the influence of the technological level of milking on work efficiency and water and electricity consumption. On the other hand, section 3.2 presents the evaluation of cows' milking, taking into account the relationship between the milking robot and the milk yield of cows.
The publication looks to me like it consists of several independent parts. The introduction and discussion are a good analysis of the literature on the subject. Material and method and chapter 3.1 are an interesting study on Polish dairy farms on energy and working time. Chapter 3.2 is a good analysis on milk yield and AMS and optimal herd size. However, I have not found the link that ties it all together into one publication.
The link connecting subsections 3.1 and 3.2 is a common concept of progress in the field of cow milking, where in one subsection (3.1) technological progress was considered, and in the second subsection (3.2) an example of technical progress.
Line 771: There is also a DOI here.
Thanks for the tip regarding DOI. I have inserted the DOI designation in the indicated article in References.
Line 773: Source misspelled. Correct: "Roboter: Zu viel Leerlauf."
Thank you for correcting the spelling of the article title. I have included the correct wording of the title in the required place in References.
The sources can be revised again altogether.
I looked through the literature on References again and made some minor adjustments.
All corrections have been made in red in the attached file with the article taking into account the changes and additions.

Round 2
Reviewer 2 Report
The revision has improved understandability.